# Mitophagy Mediates the Beige to White Transition of Human Primary Subcutaneous Adipocytes Ex Vivo

**DOI:** 10.3390/ph15030363

**Published:** 2022-03-17

**Authors:** Attila Vámos, Abhirup Shaw, Klára Varga, István Csomós, Gábor Mocsár, Zoltán Balajthy, Cecília Lányi, Zsolt Bacso, Mária Szatmári-Tóth, Endre Kristóf

**Affiliations:** 1Laboratory of Cell Biochemistry, Department of Biochemistry and Molecular Biology, Faculty of Medicine, University of Debrecen, H-4032 Debrecen, Hungary; vamos.attila@med.unideb.hu (A.V.); abhirup.shaw@med.unideb.hu (A.S.); klaravarga95@gmail.com (K.V.); balajthy@med.undeb.hu (Z.B.); szatmari-toth.maria@med.unideb.hu (M.S.-T.); 2Doctoral School of Molecular Cell and Immune Biology, University of Debrecen, H-4032 Debrecen, Hungary; 3Department of Biophysics and Cell Biology, Faculty of Medicine, University of Debrecen, H-4032 Debrecen, Hungary; csomos.istvan@med.unideb.hu (I.C.); mocsgab@med.unideb.hu (G.M.); bacso@med.unideb.hu (Z.B.); 4Laser Clinic, H-1012 Budapest, Hungary; lezerklinika@gmail.com; 5Faculty of Pharmacy, University of Debrecen, H-4032 Debrecen, Hungary

**Keywords:** obesity, beige adipocytes, mitophagy, thermogenesis, uncoupling protein 1, parkin

## Abstract

Brown and beige adipocytes have multilocular lipid droplets, express uncoupling protein (UCP) 1, and promote energy expenditure. In rodents, when the stimulus of browning subsides, parkin-dependent mitophagy is activated and dormant beige adipocytes persist. In humans, however, the molecular events during the beige to white transition have not been studied in detail. In this study, human primary subcutaneous abdominal preadipocytes were differentiated to beige for 14 days, then either the beige culture conditions were applied for an additional 14 days or it was replaced by a white medium. Control white adipocytes were differentiated by their specific cocktail for 28 days. Peroxisome proliferator-activated receptor γ-driven beige differentiation resulted in increased mitochondrial biogenesis, UCP1 expression, fragmentation, and respiration as compared to white. Morphology, UCP1 content, mitochondrial fragmentation, and basal respiration of the adipocytes that underwent transition, along with the induction of mitophagy, were similar to control white adipocytes. However, white converted beige adipocytes had a stronger responsiveness to dibutyril-cAMP, which mimics adrenergic stimulus, than the control white ones. Gene expression patterns showed that the removal of mitochondria in transitioning adipocytes may involve both parkin-dependent and -independent pathways. Preventing the entry of beige adipocytes into white transition can be a feasible way to maintain elevated thermogenesis and energy expenditure.

## 1. Introduction

The prevalence of obesity has dramatically increased worldwide in recent decades [1] and can be linked with many factors, including metabolic, genetic, environmental, and behavioral impacts [2,3]. Obesity significantly enhances the risk of many diseases, including metabolic syndrome, type 2 diabetes [4], nonalcoholic fatty liver disease [5], cardiovascular diseases, and certain types of tumors [6], which are leading causes of death [7]. Recently, obesity was recognized as an independent risk factor for mortality from severe acute respiratory syndrome coronavirus (SARS-CoV)-2 infections [8,9]. The available treatments are of diet, exercise, or lifestyle interventions and bariatric surgery [10,11]; however, effective anti-obesity therapeutic strategies are limited.

Brown adipose tissue (BAT), which contains thermogenic brown and beige adipocytes [12], is capable of energy dissipation through the production of heat [13], mainly mediated by Uncoupling protein (UCP) 1-dependent proton leak, which uncouples oxidative phosphorylation from ATP generation in the mitochondria [14]. BAT plays a key role in maintaining the constant core body temperature through nonshivering thermogenesis [15] and might open up promising therapeutic opportunities to combat obesity [16]. BAT can be found in six anatomical regions (cervical, supraclavicular, axillary, mediastinal, paraspinal, and abdominal) in adult humans, which amount to 4.3% of total fat and 1.5% of total body mass [17]. Based on mathematical predictions, BAT can oxidize around 4 kg fat per year in adult humans and its thermogenic activity can contribute up to 5% of the basal metabolic rate [18].

White adipocytes function as long-term energy storage, contain few mitochondria and a single large lipid droplet, and express a low level of UCP1 [19,20]. Brown and beige adipocytes originate from distinct precursor cells, characterized by multilocular small lipid droplets, a high mitochondrial content, and a detectable level of UCP1 expression [21,22]. Beige and white adipocytes are derived from the same mesenchymal precursors and beige cells can be found in a masked condition in subcutaneous white adipose tissue (WAT) depots [21]. Beige adipocytes contain a low amount of UCP1 in basal conditions, and have to be activated, e.g., by the cold, for thermogenesis, which is mainly mediated through β-adrenergic stimulation [23]. However, the regulation of beige adipocyte maintenance and inducibility (the process is often called “browning”) in humans has remained elusive.

Autophagy is a well-described intracellular catabolic process, in which cargoes, such as protein aggregates or damaged organelles, are delivered by double-membrane-bound structures, termed autophagosomes, to the lysosomes for degradation and their components are recycled [24,25]. Many highly conserved autophagy-related (ATG) proteins are responsible for the biogenesis of autophagosomes [26]. Upon the induction of autophagy, the cytosolic form of Microtubule-associated protein 1A/1B-light chain 3 (LC3)-I is conjugated to phosphatidylethanolamine to generate the lipidated LC3-II, which is then recruited to autophagosomal membranes. LC3 is a well-accepted autophagosome marker, and LC3-II content is an indicator of the amount of autophagosome formation [27]. Detecting the conversion of LC3-I to LC3-II by immunoblotting is a commonly used method to follow autophagy activity [28].

Damaged or unwanted mitochondria can be removed by selective autophagy, termed mitophagy, which is considered a crucial mechanism of mitochondrial quality control [29]. In the adapter-mediated, ubiquitin-dependent mitophagy pathway, mitochondrial depolarization initiates the accumulation of phosphatase and tensin homolog–induced putative kinase (PINK) 1 in the outer mitochondrial membrane, resulting in the recruitment of parkin from the cytosol, which ubiquitinates the outer mitochondrial proteins. The selective autophagy adapter proteins, such as Calcium Binding and Coiled-Coil Domain 2/Nuclear Domain 10 Protein 52 (CALCOCO2/NDP52), Optineurin (OPTN), Neighbor of BRCA1 Gene (NBR) 1, and p62 (encoded by *SQSTM1* gene) link the parkin ubiquitinated mitochondrial proteins and LC3, leading to the sequestration of mitochondria into autophagosomes [30,31]. Furthermore, the adapter-independent, ubiquitin-independent mitophagy process is characterized by direct interaction between LC3 and mitochondria-localized proteins, such as BCL2 Interacting Protein (BNIP) 3, BNIP3 Like/NIP3-Like Protein X (BNIP3L/NIX), or FUN14 Domain Containing (FUNDC) 1 [32,33]. We have considerable knowledge about selective mitophagy; however, many questions remain unanswered, such as the extent to which mitochondrial clearance is regulated in a cell type- or tissue-specific manner.

Recent publications proved the significance of autophagy in the regulation of beige adipocyte thermogenesis [34]. In rodents, Kajimura et al. showed the role of parkin-dependent mitophagy in the beige to white adipocyte transition as a result of the removal of β-adrenergic stimulus, which resulted in inactive but reactivation-capable beige adipocytes with white morphology [35,36]. However, this process has not been characterized in humans so far. Recently, our research group described how cyclic adenosine monophosphate (cAMP)-driven thermogenic activation regulates mitophagy in human masked and mature beige adipocytes ex vivo. Our data indicated continuous mitochondrial clearance in these adipocytes, which was rapidly repressed in response to short-term adrenergic stimulus, pointing to a fast regulatory mechanism to provide high mitochondrial content for thermogenesis [37].

In this study, we follow up the autonomous transition of human primary subcutaneous abdominal beige adipocytes to white and investigate whether mitophagy is activated during this process. The transition, which involved both parkin-dependent and independent pathways, resulted in characteristic gene expression, morphological, and functional features of the white adipocytes; however, the converted cells could be strongly activated by a cell-permeable cAMP analogue. Our results translate previously obtained rodent data to a human ex vivo system, which provides a model for further research to characterize beige to white transitioned human adipocytes in the near future.

## 2. Results

### 2.1. Thermogenic Competency of Human Abdominal Subcutaneous Derived Adipocytes Is Induced following Continuous Peroxisome Proliferator-Activated Receptor (PPAR) γ Stimulation and Subsides as a Result of Beige to White Transition

To study the adipogenic potential of primary human adipose-derived stromal cells (hASCs) and the thermogenic competency of differentiated adipocytes, our research group optimized previously published white [38] and brown/beige [39] adipogenic differentiation protocols. These regimens contain diverse compositions of hormones [40,41], in which the PPARγ agonist rosiglitazone is the key driver of browning [42]. As expected, abdominal subcutaneous hASCs expressed the major functional marker gene and protein of thermogenesis, UCP1, at the limit of detection. Moderate UCP1 expression was found in adipocytes that were differentiated up to 28 days to white ex vivo (Figure 1A,B). Consistent with previous results, continuous PPARγ stimulation resulted in a marked increase in gene and protein expression of UCP1 in adipocytes differentiated to beige compared with white ones. UCP1 was further upregulated when beige differentiation was carried out for three or four weeks. When the beige cocktail was replaced by the white and rosiglitazone was omitted at the fourteenth day of differentiation, UCP1 gene and protein expression tended to elevate in the following week, similarly to those adipocytes that were continuously exposed to the beige regimen. After two weeks of rosiglitazone withdrawal, UCP1 gene and protein expression was significantly decreased as compared to beige adipocytes (Figure 1A,B) and showed a gene expression level that was comparable to white adipocytes (Figure 1A). The decline in UCP1 expression was slower at the protein level (Figure 1B) than at the mRNA level (Figure 1A). The expression of another thermogenic marker gene, Cell Death Inducing DFFA-Like Effector A (*CIDEA*) [43] (Figure 1C), followed the pattern of *UCP1* mRNA expression (Figure 1A). In contrast, the *LEP* gene, which encodes the white adipokine, leptin [44], was expressed at a low level in both preadipocytes and beige adipocytes; however, it was strongly upregulated when adipocytes were constantly differentiated in the presence of the white cocktail or when the beige regimen was replaced by the white (Figure 1D).

As expected, white adipocytes gained a few large lipid droplets in a time-dependent manner. By contrast, beige differentiation resulted in numerous smaller droplets. However, the droplets merged and enlarged when the beige protocol was replaced by the white (Figure 2A). The quantification of this phenomenon was performed by texture sum variance, which separated the white and beige adipocyte populations well in several previous cellular models [40,41,45,46], is shown in Figure 2B. To investigate the mitochondrial morphology, we performed immunostaining for translocase of outer mitochondrial membrane (TOM) 20 [37,47] (Figure 2A; see the secondary antibody control in Appendix A). Consistent with the literature, the number of fragmented mitochondria that reflect thermogenic potential [48] was higher during beige compared to white differentiation. However, the number of fragmented mitochondria was decreased to the same as that of the white adipocytes within a week when the beige regimen was discontinued, followed by the white (Figure 2A,C). The changes in UCP1 content and morphological features of the adipocytes suggest that they strongly increase their thermogenic competency, for up to four weeks, as a result of PPARγ agonist; however, they could undergo beige to white transition in response to the removal of the browning inducer.

### 2.2. Elevated Mitochondrial Content, Respiration, and Extracellular Acidification of Beige Adipocytes Disappear after Their Transition to the White Phenotype

As a next step, we investigated the mitochondrial content and function during differentiation and transition. As expected, the mitochondrial DNA (mtDNA) number was higher in beige compared to white differentiated adipocytes and subsequently decreased as a result of beige to white transition (Figure 3A). White and beige differentiation resulted in the same expression level of the mitochondrial biogenesis master regulator, PPARγ coactivator (PGC) 1α [49]. In response to transition, *PGC1α* tended to be downregulated; however, this effect did not reach the level of statistical significance. Undifferentiated progenitors expressed *PGC1α* only at the limit of detection (Figure 3B).

Then we carried out an extracellular flux analysis to reveal the functional parameters of differentiated adipocytes [41,50,51]. In agreement with previously reported data [40,51], the basal oxygen consumption rate (OCR) of beige adipocytes was higher than that of white ones. Adipocytes undergoing two weeks of transition had significantly higher basal OCR than the white, but lower than the beige adipocytes, which were differentiated for the same period of time. As expected, the cell-permeable cAMP analogue, which mimics adrenergic stimulus-driven activation of thermogenesis, promptly increased the OCR of each type of adipocytes. Proton leak respiration of cAMP-stimulated adipocytes, which positively correlates with UCP1 activity, could be assessed after the inhibition of the ATP synthase complex by oligomycin [50,52]. Beige adipocytes had elevated stimulated and proton leak OCR compared to the white ones. After the transition, these parameters remained comparable to those observed in beige adipocytes (Figure 3C). Consistent with previous results [41,51], the basal extracellular acidification rate (ECAR) was increased in beige compared to white adipocytes, while the transition had a significant suppressing effect on this parameter. Although cAMP stimulated the ECAR of each adipocyte type, fully differentiated and converted beige adipocytes showed a greater response to the thermogenic cue than white ones (Figure 3D). In summary, our data suggest that adipocytes that undergo the beige to white transition appear as a cell population with distinct features as compared to fully differentiated white or beige cells.

### 2.3. Autophagy Is Increased at Beige to White Transition

In rodents, it was proven that the withdrawal of cold or β3-adrenergic stimuli activates mitophagy and mediates the beige to white transition in vivo [35]. Primarily, we investigated the expression of *ATG* genes, which orchestrate autophagosome formation. In response to transition, *ATG5* tended to be upregulated in the second week; however, this difference did not reach statistical significance (Figure 4A). The mRNA expression of *ATG7* (Figure 4B) and *ATG12* (Figure 4C) was significantly increased in the first week of transition compared to beige adipocytes. The ATG genes investigated were expressed less in undifferentiated preadipocytes and at a greater rate, but to the same extent, in white and beige adipocytes (Figure 4A–C).

To assess the ongoing autophagy, we examined the specific autophagy marker, the conversion of LC3-I to LC3-II, in adipocytes that were fully differentiated to white or beige or underwent transition. Quantification of this process by immunoblotting is a widely accepted method to monitor autophagy rate [53]. We found a continuous elevation of the LC3-II/LC3-I ratio in white adipocytes differentiated up to 28 days, which indicated high autophagy activity. In beige adipocytes, the activity remained at a moderate level, significantly lower than with the white cells, after four weeks of differentiation. When the beige protocol was replaced by white, the autophagy level was significantly increased after two weeks, as compared to the fully differentiated beige adipocytes (Figure 4D). This was confirmed when the subcellular distribution of LC3 was visualized by immunostaining (Figure 5A; see secondary antibody control in Appendix A). As expected, white adipocytes, which were differentiated after three or four weeks, contained more LC3 punctae per cell than beige ones. In addition, the beige to white transition significantly increased the number of LC3 punctae, as compared to the beige adipocytes (Figure 5B). Our data demonstrated that general autophagy was induced in a cell-autonomous manner during the ex vivo beige to white transition of human subcutaneous adipocytes.

### 2.4. Beige Differentiation Represses, while Transition to White Increases, Mitophagy Involving Selective Autophagy Adapters

We performed co-immunostaining of TOM20 and LC3 (Figure 5A; see secondary antibody control in Appendix A) to follow autophagosome formation and the delivery of mitochondria for degradation [47]. Then we quantified the colocalization of the autophagosome and mitochondrial markers using Pearson correlation coefficient (PCC) values. Consistent with the increased autophagy activity (Figure 4), we found elevated PCC values during white adipogenesis for four weeks compared with beige adipocytes. The colocalization was stronger in adipocytes that underwent transition than in fully differentiated beige cells (Figure 5C).

Parkin, an E3 ubiquitin ligase encoded by the *PARK2* gene, is one of the key regulators of mitophagy [54]. Parkin was expressed at a low extent in preadipocytes at both the mRNA (Figure 6A) and protein levels (Figure 6B). The applied adipogenic protocols similarly upregulated its expression. The transition did not alter the expression of parkin at the gene (Figure 6A) or protein levels (Figure 6B). Next, we investigated the abundance of selective autophagy adapter proteins that are consumed, building a molecular link between the target organelles and LC3-II of the autophagosomes during ongoing mitophagy [55]. The NBR1 protein content of white adipocytes was comparable to that of the undifferentiated progenitors and showed a decreasing trend after up to 28 days of differentiation (Figure 6C). The p62 protein was detectable at constant, moderate levels in preadipocytes and white adipocytes (Figure 6D). In beige adipocytes, significantly more of the aforementioned adapters could be detected. This further supports the low activity of selective autophagy in beige adipocytes. After the removal of the browning inducer from differentiation media, the NBR1 (Figure 6C) and p62 (Figure 6D) amount declined significantly after one or two weeks, respectively. Our data suggest that the selective autophagic degradation of mitochondria is enhanced during the beige to white transition.

Next, we assessed the expression of other marker genes related to the adapter and parkin-dependent mitophagy pathway, *OPTN* (Figure 6E) and *CALCOCO2/NDP52* (Figure 6F). Both genes were expressed at a low level in preadipocytes. *OPTN* was expressed to the same extent in white and beige adipocytes and tended to be upregulated in a time-dependent manner, but the transition did not influence the mRNA level (Figure 6E). *CALCOCO2/NDP52* expression was increased during white adipogenesis and transition compared with fully differentiated beige adipocytes (Figure 6F); this suggests the possibility of enhanced removal of the mitochondrial mass by the NDP52-dependent pathway.

### 2.5. Parkin-Independent Mitophagy-Related Genes Are Induced during Transition

Finally, to study whether parkin-independent mitophagy contributes to beige to white transition, we investigated the expression of several genes that are involved in this pathway, *FUNDC1*, *BNIP3*, *BNIP3L/NIX*, FKBP Prolyl Isomerase 8 (*FKBP8*), and BCL2 Like 13 (*BCL2L13*) (Figure 7). The aforementioned markers were expressed to a low extent in undifferentiated progenitors. *BNIP3L/NIX*, *FKBP8*, and *BCL2L13* expression tended to increase during white adipogenesis, in a time-dependent manner. Similar expression levels of *FUNDC1* and *BNIP3* were found during white or beige differentiation. After four weeks of differentiation, the expression of *BNIP3L/NIX*, *FKBP8*, and *BCL2L13* was repressed in beige compared to white adipocytes. Two weeks following the replacement of a beige protocol with white, the investigated parkin-independent mitophagy markers were significantly upregulated (Figure 7). This suggests that the parkin-independent pathway plays an important role during the beige to white transition of human subcutaneous adipocytes.

## 3. Discussion

BAT plays a central role in the energy homeostasis of mammals that are constantly exposed to cold challenge [13]. Following the detection of active BAT depots by nuclear imaging approaches in adult humans [56,57,58], a strong negative correlation between obesity and the amount of active BAT was revealed [59,60]. Independent studies have suggested that BAT depots in adult humans are predominantly composed of beige cells [12,61,62]. Transplants of human beige adipocytes improved diet-induced obesity and systemic metabolism in mice, which highlights the possibility of the therapeutic application of beige cell implantation in the treatment of obesity and metabolic syndrome [63,64]. This inspired researchers to characterize beige adipogenesis and thermogenic activation in distinct human cellular models. To our knowledge, however, the majority of these ex vivo studies have covered the differentiation period for a maximum of two weeks. Of note, the potential application of beige adipocyte activation or transplantation assumes that the applied cells maintain their energy expenditure for a significant period of time. Although abdominal subcutaneous WAT of human adults is not highly enriched in thermogenic adipocytes [17], it contains progenitors that can give rise to beige cells [65,66]. Due to its relative accessibility, hASCs isolated from stromal–vascular fractions (SVFs) of abdominal subcutaneous fat biopsies or aspirations are frequently used for research and in regenerative medicine [67].

In this study, we followed the white and beige adipocyte differentiation of primary abdominal subcutaneous-derived hASCs for four weeks. The extension of the PPARγ-driven beige differentiation resulted in further upregulation of UCP1, both at the mRNA (Figure 1A) and protein (Figure 1B) level, and the *CIDEA* gene (Figure 1C), while in white adipocytes they were expressed constantly at a moderate level. This phenomenon was reproduced by our research group in Simpson–Golabi–Behmel syndrome (SGBS) adipocytes [41], a cell line that is an accepted and widely used model of human white and beige adipogenesis [68,69]. Consistent with our previous findings [40,41], the size and locularity of lipid droplets were different in the two cell populations during the entire differentiation (Figure 2A,B).

In beige and brown adipocytes, mitochondria are critical for thermogenesis and energy metabolism. Mitochondria are fragmented in response to an adrenergic cue in rodent brown adipocytes, contributing to support uncoupled respiration and enhanced energy expenditure [70,71]. Pisani et al. have shown that UCP1-positive human adipocytes contain mitochondria mostly with a fragmented morphology [48]. Recently, we found that cAMP-driven thermogenic stimulation resulted in increased mitochondrial fragmentation in human masked and mature beige adipocytes, which were differentiated from the same progenitor populations for two instead of four weeks [37]. When we sustained the beige differentiation, more mitochondria were fragmented in contrast to white adipocytes, in which these dynamic organelles remained elongated (Figure 2C).

The regulation of mitochondrial biogenesis and clearance is important for energy homeostasis and maintaining the optimal number of mitochondria [32]. Mitochondrial biogenesis is controlled by several nuclear-coded transcriptional regulators, such as PGC1α [72]. Although the expression of *PGC1α* at mRNA tended to be elevated in the early phase of beige differentiation, it did not differ between white and beige adipocytes to a statistically significant degree (Figure 3A). However, the amount of mtDNA showed an increasing tendency during the long-term beige adipogenesis (Figure 3B). The functional extracellular flux assay detected high basal, cAMP-stimulated, and proton leak OCR and more prominent extracellular acidification, both in basal and activated conditions, in the case of beige adipocytes that were differentiated in the presence of rosiglitazone for four weeks. Respiration and extracellular acidification were significantly repressed in white adipocytes; however, they could be effectively stimulated in response to the cell-permeable cAMP analogue, suggesting that some of the adipocytes that were differentiated in the presence of the white cocktail are masked beige cells (Figure 3C,D).

The activation of beige cells, for example by cold exposure or β-adrenergic receptor agonists, is required for maintaining high UCP1 expression. Cold exposure induces norepinephrine release from the sympathetic nervous system, which binds the β3-adrenergic receptor [13]. The β-adrenergic signaling cascade is mediated through adenylate cyclase activation by Gs proteins, which leads to the production of cAMP for protein kinase A (PKA) activation. This promotes lipolysis, the breakdown of triglycerides, resulting in free fatty acid release that activates UCP1 [23]. PKA has several downstream targets, including cAMP Response Element Binding Protein (CREB), members from MAP kinase pathways (Erk1/2, p38, JNK), and hormone-sensitive lipase (HSL), which facilitate the enhancement of thermogenesis by upregulating thermogenic gene expression and/or mobilizing substrates to fuel thermogenesis [73,74].

PPARγ is well known as a master regulator of both white and brown adipocyte differentiation. Some synthetic PPARγ agonists, such as rosiglitazone, function as inducers of the beige fat gene transcription program in white adipocytes, which is mediated by SIRT1, PRDM16, C/EBPa, and PGC1α-dependent mechanisms [75]. PPARγ directly regulates the expression of many genes, which are involved in the central functions of adipocytes, such as lipid transport, lipid metabolism, insulin signaling, and adipokine production [76]. PPARγ target genes can control processes such as lipid transport (*FABP4*), fatty acid uptake (*LPL*, *FATP/SLC27A1*, *OLR1*), the recycling of intracellular fatty acids (*PEPCK/PCK1*, *GK*, *AQP7*) [77], and lipolysis (*GPR81*) [78]. In addition, PPARγ regulates insulin sensitivity via the expression of adipokines [79].

Elevated mitochondrial fragmentation, mtDNA, and OCR raise the possibility of suppressed mitophagy in beige adipocytes. When the same progenitors were differentiated for two weeks, irrespective of the applied protocol, a few hours of cAMP treatment not only upregulated thermogenesis-related genes but also quickly downregulated mitophagy via PKA, resulting in more mitochondria and increased UCP1 levels [37]. In a long-term differentiation setting, sustained rosiglitazone administration also resulted in a moderate suppression of mitophagy, shown by attenuated LC3-I to LC3-II conversion (Figure 4D), the appearance of LC3-positive punctae (Figure 5B), colocalization of punctae and mitochondria (Figure 5C), and the degradation of adapter proteins (Figure 6C,D). More research is needed to explore the underlying molecular mechanisms of how mitophagy is kept at a moderately low level in beige adipocytes.

Altshuler-Keylin et al. induced beigeing of WAT in male mice by intraperitoneal administration of the β3-adrenergic agonist, CL 316,243 for seven consecutive days. Low autophagy activity was observed in newly differentiated beige adipocytes [35], which is consistent with the ex vivo results presented here (Figure 4 and Figure 5). After the withdrawal of this stimulus, they found that beige adipocytes lost their morphological and thermogenic characteristics and were converted to “white-like” adipocytes, triggered by mitochondrial clearance via mitophagy. During the beige to white adipocyte transition, the expression of autophagy-related *ATG5* and *ATG12* genes was upregulated, the number of Green Fluorescent Protein (GFP)-LC3 punctae and the colocalization of GFP-LC3 and TOM20 were significantly increased, the protein level of LC3-II was elevated, and in parallel the selective autophagy adapter proteins, NBR1 and p62, were degraded, compared to in mice that were chronically treated with the β3-adrenergic agonist during the entire experimental period [35].

In our experiments, we applied the human primary subcutaneous abdominal derived adipocyte ex vivo model to characterize the beige to white transition process in the context of mitophagy. UCP1 (Figure 1A,B), *CIDEA* (Figure 1C), and *LEP* (Figure 1D) expression, the morphology of the lipid droplets (Figure 2A,B), mitochondrial fragmentation (Figure 2A,C), and the basal respiration (Figure 3C) of the adipocytes were significantly altered as a result of the transition; therefore, these cells gained several features characteristic of white adipocytes. A similar phenomenon was observed recently when we carried out a 28-day differentiation, in parallel with the replacement of beige protocol to white on the 14th day in the SGBS preadipocyte line [41]. In contrast to our previous observations in SGBS adipocytes and the data presented here, Guennoun et al. observed a temporarily high UCP1 content of 14-day-differentiated SGBS adipocytes, even in response to the white differentiation protocol in the absence of any browning inducers. Interestingly, when the white differentiation of SGBS cells was extended for two additional weeks, the expression of UCP1 significantly decreased [80]. The contribution of autophagy to this surprising finding has not been investigated so far.

Our data, shown in Figure 4 and Figure 5, are consistent with the results of the in vivo study by Kajimura’s group [35] and suggest that the autophagy pathway is activated for the clearance of beige adipocyte mitochondria during the adaptive transition induced by the removal of the browning inducer, thereby regulating the entry of beige adipocytes into a thermogenically inactive dormant state. However, beige adipocytes that underwent transition ex vivo responded more effectively to the adrenergic stimulus mimicking dibutyril-cAMP by the activation of OCR, proton leak respiration (Figure 3C), and ECAR (Figure 3D) than the white ones. Of note, a significant amount of UCP1 protein remained expressed after two weeks of transition (Figure 1B), which could underlie why converted adipocytes had higher stimulated and proton leak OCR compared to white ones. The observed functional differences suggest that converted beige and white adipocytes can be classified into two distinct cell populations. Systematic studies are needed to further explore the molecular signatures of the thermogenically active beige, converted beige, and white adipocytes in humans.

As a final step, we sought to clarify how the selective removal of mitochondria is mediated during the transition process. In the literature, contradictory data have been published about the involvement of parkin in the maintenance of murine beige adipocytes. During the adipogenesis of 3T3-L1 adipocytes, increased parkin expression was observed, whereas its expression decreased as a result of rosiglitazone treatment [81]. Lu et al. found that parkin expression was induced during mouse beige adipocyte differentiation; moreover, they demonstrated the retention of mitochondria-rich beige adipocytes even after the elimination of adrenergic stimuli in *PARK2* knockout mice [36]. In contrast to these findings, Corsa and colleagues found that parkin deletion in mouse adipocytes did not affect adipogenesis, beige to white transition, and the maintenance of beige adipocytes [82]. Recently, we reported that parkin-dependent and -independent mitophagy-associated genes were expressed in human masked and mature beige adipocytes, and the cAMP-driven thermogenic stimulus resulted in decreased expression of parkin-dependent mitophagy-related genes [37]. In the current study, we have found that the gene and protein expression of parkin was not affected during beige to white transition (Figure 6A,B). However, the level of selective adapter proteins, NBR1 (Figure 6C) and p62 (Figure 6D) significantly decreased and the expression of *CALCOCO2/NDP52* (Figure 6F) and the investigated parkin-independent mitophagy-related genes (Figure 7) was significantly elevated during transition, compared to fully differentiated beige adipocytes. In summary, our data suggest that both parkin-dependent and -independent mitophagy pathways are involved in the regulation of mitochondrial elimination during beige to white adipocyte transition.

p62 is a multifunctional protein involved in several signaling pathways affecting various cellular processes, such as inflammation, cell death, tumorigenesis, and metabolism [83,84]. It has been reported that whole-body p62 knockout mice show an obese phenotype due to increased adiposity and reduced energy expenditure [85]. Furthermore, the mitochondrial function of BAT in adipocyte-specific p62^−/−^ mice was impaired, resulting in BAT becoming unresponsive to β3-adrenergic stimuli [86]. This suggests that p62 plays a significant role in the regulation of thermogenesis in BAT. A recent paper has demonstrated that NBR1 is required for the repression of adaptive thermogenesis via decreasing the activity of PPARγ in BAT of p62-deficient mice; thereby, the inhibitory role of NBR1 in thermogenesis in the presence of p62 inactivation was identified [87]. Based on these studies, further investigations may reveal the role of p62 and NBR1 in the thermogenesis of human brown and beige adipocytes.

Individuals with obesity possess less active BAT but more “brownable” fat than lean ones [17]. These “brownable” depots might contain a large number of beige adipocytes undergoing transition, in which autophagy and mitophagy are highly active. This is supported by the fact that *ATG* and autophagosome-related genes are highly expressed in the visceral and subcutaneous WAT of patients with obesity [88,89]. In the future, well-established molecular markers and histological or cell sorting methods are necessary to discriminate between white, active beige, and dormant beige adipocytes in distinct anatomical areas. This might allow researchers to analyze gene expression changes during conversion in individual cells, which might reveal the novel molecular targets that control this process. More investigations are needed to use the inhibition of autophagy to cure diseases but there are feasible findings that show practicable perspectives. For example, García-Pérez et al. summarized that targeting autophagy in the early stage of SARS-CoV-2 infections can be a potential therapeutic strategy against viral replication and in the regulation of the exacerbated inflammatory response [90]. A better understanding of the key molecular events that determine the entry into beige to white transition may offer new opportunities for specifically preventing this process in order to maintain active heat-producing adipocytes; they may be pharmacologically activated or transplanted, for instance, in humans for improving energy metabolism and combatting obesity.

## 4. Materials and Methods

### 4.1. Materials

All chemicals were obtained from Sigma Aldrich (Munich, Germany) unless stated otherwise.

### 4.2. Isolation, Maintenance, and Differentiation of hASCs

hASCs were obtained and isolated from SVFs of human subcutaneous abdominal adipose tissue of healthy donors undergoing planned surgeries, as previously described [37,40]. The absence of mycoplasma was ascertained by a PCR Mycoplasma kit (PromoCell, Heidelberg, Germany). hASCs were seeded in six-well plates (Costar, Corning, NY, USA) and differentiated for 14, 21, and 28 days as indicated, following the white [37,40,41] or beige [37,39,40,41] adipogenic differentiation protocols. White differentiation was initiated by using a serum-free DMEM-F12 medium supplemented with 33 μM biotin, 17 μM pantothenic acid, 100 U/mL penicillin/streptomycin, 10 μg/mL human apo-transferrin, 20 nM human insulin, 200 pM triiodothyronine, 100 nM cortisol, 2 μM rosiglitazone (Cayman Chemicals, Ann Arbor, MI, USA), 25 nM dexamethasone, and 500 μM 3-isobutyl-1-methylxanthine (IBMX). Four days later, rosiglitazone, dexamethasone, and IBMX were removed. The beige protocol was initiated for four days, applying serum-free DMEM-F12 containing 33 μM biotin, 17 μM pantothenic acid, 100 U/mL penicillin/streptomycin, 10 μg/mL apo-transferrin, 0.85 μM human insulin, 200 pM triiodothyronine, 1 μM dexamethasone, and 500 μM IBMX. Then 500 nM rosiglitazone was added to the cocktail, while dexamethasone and IBMX were omitted. Where indicated, after a 14-day beige differentiation period, the transition was made to a white differentiation protocol for a further seven or 14 days, respectively. The transition was initiated by the addition of 100 nM cortisol and the removal of 500 nM rosiglitazone, the key driver of beige differentiation. The concentration of human insulin decreased by 42.5-fold at the induction of transition. Cells were incubated at 5% CO_2_ and 37 °C, and media were replaced at an interval of four days.

### 4.3. Nucleic Acid Isolation, RT-PCR, and qPCR

Cells were collected in Trizol reagent (Thermo Fisher Scientific, Waltham, MA, USA), followed by manual isolation of RNA and DNA by chloroform extraction and isopropanol or ethanol precipitation, respectively. RNA quality was evaluated by Nanodrop (Thermo Fisher Scientific), and cDNA was generated by a TaqMan reverse transcription reagent kit (Thermo Fisher Scientific) followed by qPCR analysis [37,91]. Gene expression was normalized to *GAPDH*. A list of all the probes is provided in Appendix A. Quantification of mtDNA was performed by qPCR as previously described [37,46].

### 4.4. Antibodies and Immunoblotting

Sample separation was performed by SDS-PAGE, followed by transfer to a PVDF membrane. The membrane was blocked by 5% skimmed milk solution [37,91]. The following primary antibodies were used: anti-UCP1 (1:750, R&D Systems, Minneapolis, MN, USA, MAB6158), anti-p62 (1:5000, Novus Biologicals, Centennial, CO, USA, NBP1-49956), anti-LC3 (1:2000, Novus Biologicals, NB100-2220), anti-Parkin (1:750, Santa Cruz Biotechnology, Dallas, TX, USA, sc-32282), anti-NBR1 (1:1000, Novus Biologicals, NBP1-71703), and anti-β-actin (1:5000, A2066). HRP-conjugated goat anti-rabbit (1:10,000, Advansta, San Jose, CA, USA, R-05072-500) or anti-mouse (1:5000, Advansta, R-05071-500) IgG was used as secondary antibodies. Immunoreactive proteins were visualized, followed by densitometry by FIJI ImageJ software (National Institutes of Health (NIH), Bethesda, MD, USA) as previously described [37].

### 4.5. Immunostaining and Image Analysis

hASCs were plated and differentiated in eight-well Ibidi μ-chambers (Ibidi GmbH, Gräfelfing, Germany) for the indicated number of days. Cells were washed once with PBS and fixed by 4% paraformaldehyde, followed by permeabilization with 0.1% saponin and blocking with 5% skimmed milk [37]. Primary antibody incubations were kept overnight with anti-TOM20 (1:75, WH0009804M1) and anti-LC3 (1:200, Novus Biologicals, NB100-2220). Secondary antibody incubation was for 3 h with Alexa Fluor 647 goat anti-mouse IgG (1:1000, Thermo Fisher Scientific, A21236) and Alexa Fluor 488 goat anti-rabbit IgG (1:1000, Thermo Fisher Scientific, A11034). Propidium iodide (PI, 1.5 μg/mL, 1 h) was used for nuclei labeling. Images were obtained with an Olympus FluoView 1000 (Olympus Scientific Solutions, Tokyo, Japan) confocal microscope and FluoView10-ASW (Olympus Scientific Solutions) software version 3.0, as previously described [37,46]. For the excitation of Alexa Fluor 488, the 488 nm line of an argon ion laser was used, while for Alexa Fluor 647, a 633-nm He–Ne laser was used; for PI, a 543-nm He–Ne laser was used. The fluorescence emissions of Alexa Fluor 488 and Alexa Fluor 647 were detected through 500–530 nm and 655–755 nm bandpass filters, respectively, while detection of the fluorescence of PI was achieved with a 555–625 nm bandpass filter. Images were taken in sequential mode to minimize crosstalk between the channels. Images of approximately 1-μm-thick optical sections, each containing 512 × 512 pixels (pixel size: ~137 nm), were obtained with a 60× UPLSAPO oil immersion objective (NA 1.35).

LC3 and TOM20 immunostaining images were converted to binary form, followed by processing with FIJI. The LC3 punctae count was determined by size (pixel^2^) 50–infinity AU with circularity 0–1 AU. Fragmented mitochondria were analyzed from the binary TOM20 immunostaining images with size (pixel^2^) 0–100 AU and circularity 0–1 AU. The optimum size values for the LC3 punctae and fragmented mitochondria were determined based on an analysis of all immunostaining images and manual verification of the counting accuracy by checking the outlines of counts. Both LC3 punctae and fragmented mitochondria content were normalized to per nucleus for individual images [37,46]. Colocalization of LC3 and TOM20 was evaluated by calculation of the PCC [37]. Texture sum variance was calculated using iCys companion software (iNovator Application Development Toolkit version 7.0, CompuCyte Corporation, Westwood, MA, USA), Cell Profiler, and Cell Profiler Analyst (The Broad Institute of MIT and Harvard, Cambridge, MA, USA) as previously described [40,45].

### 4.6. Determination of Cellular OCR and ECAR

Cells were seeded and differentiated on XF96 assay plates (Seahorse Biosciences, North Billerica, MA, USA) and differentiated for 28 days with white, beige, or beige to white transition protocols, followed by the measurement of OCR and ECAR with XF96 oximeter (Seahorse Biosciences). cAMP-stimulated OCR, ECAR, and stimulated proton leak OCR were measured as previously described [51]. Antimycin A (10 μM) was used for baseline correction. The OCR was normalized to protein content.

### 4.7. Statistics and Figure Preparation

All measured values are expressed as mean ± SD for the number of independent repetitions indicated. The normality of the distribution of data was tested by the Kolmogorov–Smirnov (*n* = 5 or more) or Shapiro–Wilk (*n* = 4) test. One-way ANOVA with a Tukey’s post hoc test was used for multiple comparisons of groups when the dataset followed normal distribution. Friedman’s test and Dunn’s multiple comparison test were used for multiple comparisons of groups when the dataset did not follow normal distribution. GraphPad Prism 9 (GraphPad Software, San Diego, CA, USA) was used for figure preparation and statistics.

## 5. Conclusions

Although the concept of beige to white adipocyte transition is not novel, the underlying transcriptional and cell biological changes have not been investigated in human primary cell models so far. We successfully modeled the white and beige differentiation of human abdominal subcutaneous adipocytes for up to 28 days and observed the beige to white transition ex vivo for the first time to our knowledge. Beige adipocytes had elevated mitochondrial biogenesis, UCP1 expression, fragmentation, and oxygen consumption compared to white adipocytes. In adipocytes that underwent the beige to white transition, these parameters were similar to those observed in white cells. During the transition, both parkin-dependent and -independent mitophagy marker genes were induced. The direct functions of individual elements of the mitophagy machinery need to be investigated through experiments involving pharmacological inhibitors and gene silencing, deletion, or overexpression. In our opinion, inhibiting the transition of beige to white adipocytes may be a way to maintain their high energy expenditure. This could potentially lead to more active BAT instead of “brownable” fat. An in-depth understanding of the molecular mechanisms behind this process could help to develop new methods to treat obesity in the future.

## Figures and Tables

**Figure 1 pharmaceuticals-15-00363-f001:**
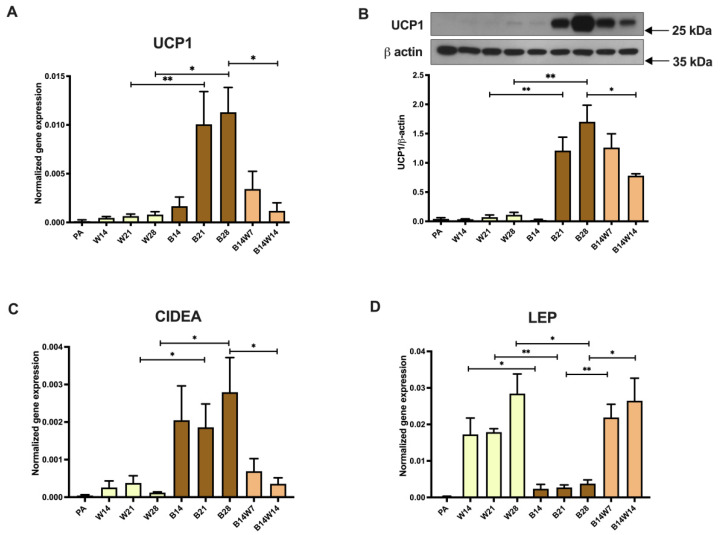
Expression of thermogenic markers was elevated following long-term rosiglitazone treatment and subsided upon beige (B) to white (W) transition. Human primary abdominal subcutaneous preadipocytes were differentiated to B for 14 days (B14), then either the B culture conditions were applied for an additional 14 days (B21 and B28) or were replaced by a W differentiation medium (B14W7 and B14W14). As a negative control, W adipocytes were differentiated by their specific cocktail (W14, W21, and W28). UCP1 (**A**) gene, (**B**) protein, (**C**) *CIDEA*, and (**D**) *LEP* gene expression. Gene expression was normalized to *GAPDH* and protein expression to β-actin. Data are presented as mean ± SD. *n* = 6. * *p* < 0.05, ** *p* < 0.01. We carried out statistics using Friedman’s test with Dunn’s multiple comparison test (**A**) or one-way ANOVA with Tukey’s post hoc test (**B**–**D**).

**Figure 2 pharmaceuticals-15-00363-f002:**
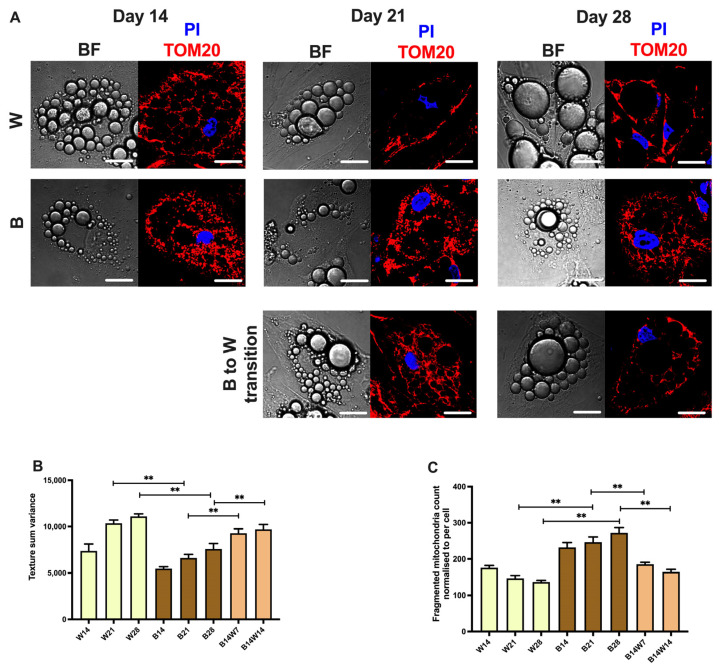
Mitochondrial fragmentation was elevated following long-term rosiglitazone treatment and subsided upon beige (B) to white (W) transition. Human primary abdominal subcutaneous preadipocytes were differentiated as in Figure 1. (**A**) Representative confocal microscopy images of TOM20 immunostaining; nuclei are labeled with propidium iodide (PI), and BF represents brightfield image; scalebars represent 10 μm. (**B**) Texture sum variance quantified from BF images. (**C**) Fragmented mitochondrial content quantified based on TOM20 immunostaining normalized to per cell. Data are presented as mean ± SD. *n* = 50 cells from three donors. ** *p* < 0.01. We carried out statistics using one-way ANOVA with Tukey’s post hoc test (**B**) or Friedman’s test with Dunn’s multiple comparison test (**C**).

**Figure 3 pharmaceuticals-15-00363-f003:**
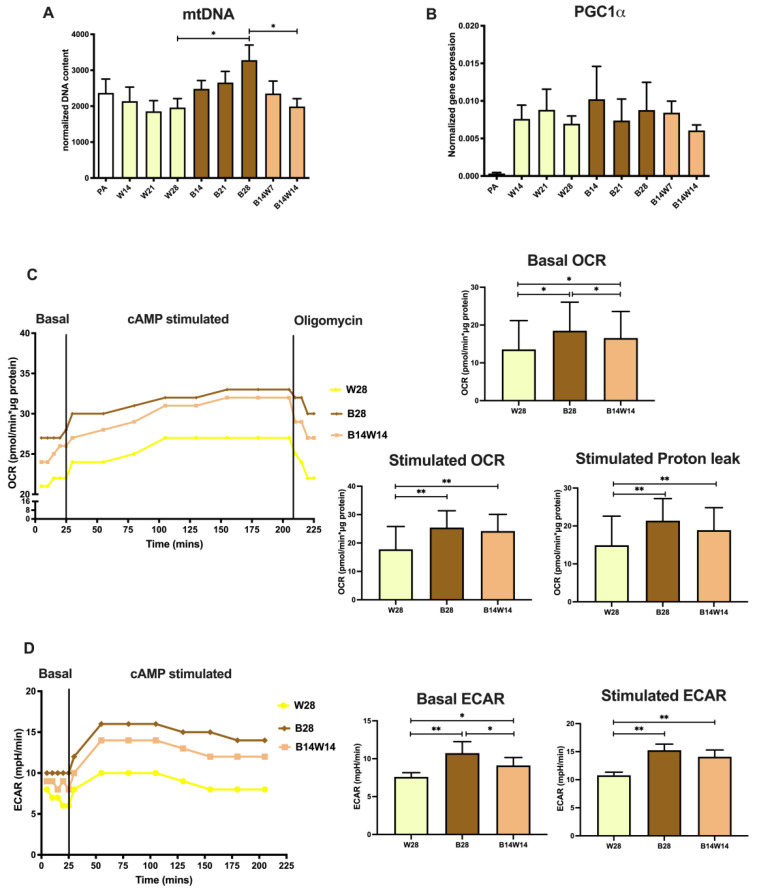
Functional parameters of mitochondrial biogenesis and thermogenesis in white (W) and beige (B) adipocytes, and in response to transition. Human primary abdominal subcutaneous preadipocytes were differentiated as in Figure 1 and Figure 2. Quantification of (**A**) total mitochondrial DNA content (*n* = 6) and (**B**) *PGC1α* gene expression (*n* = 6). (**C**) Representative oxygen consumption rate (OCR) curve, followed by quantification of basal, stimulated, and stimulated proton leak OCR (*n* = 4). (**D**) Representative extracellular acidification rate (ECAR) curve, followed by quantification of basal and stimulated ECAR (*n* = 4). Gene expression was normalized to *GAPDH*. Data are presented as mean ± SD. * *p* < 0.05, ** *p* < 0.01. We used one-way ANOVA with Tukey’s post hoc test.

**Figure 4 pharmaceuticals-15-00363-f004:**
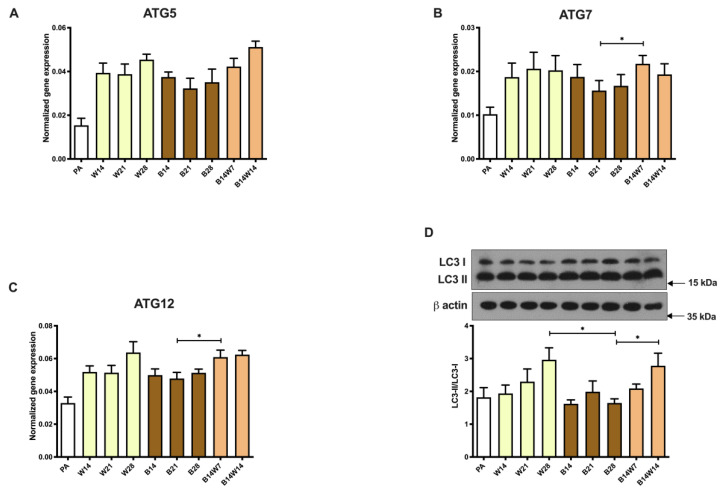
General autophagy markers showed a decreasing trend following long-term rosiglitazone treatment and were increased upon beige (B) to white (W) transition. Human primary abdominal subcutaneous preadipocytes were differentiated as in Figure 1, Figure 2 and Figure 3. (**A**–**C**) Quantification of *ATG5*, *ATG7*, and *ATG12* gene expression; (**D**) representative immunoblot and densitometry analysis of LC3-II/LC3-I protein ratio. Gene expressions were normalized to *GAPDH*. Data are presented as mean ± SD. *n* = 6. * *p* < 0.05. We used one-way ANOVA with Tukey’s post hoc test.

**Figure 5 pharmaceuticals-15-00363-f005:**
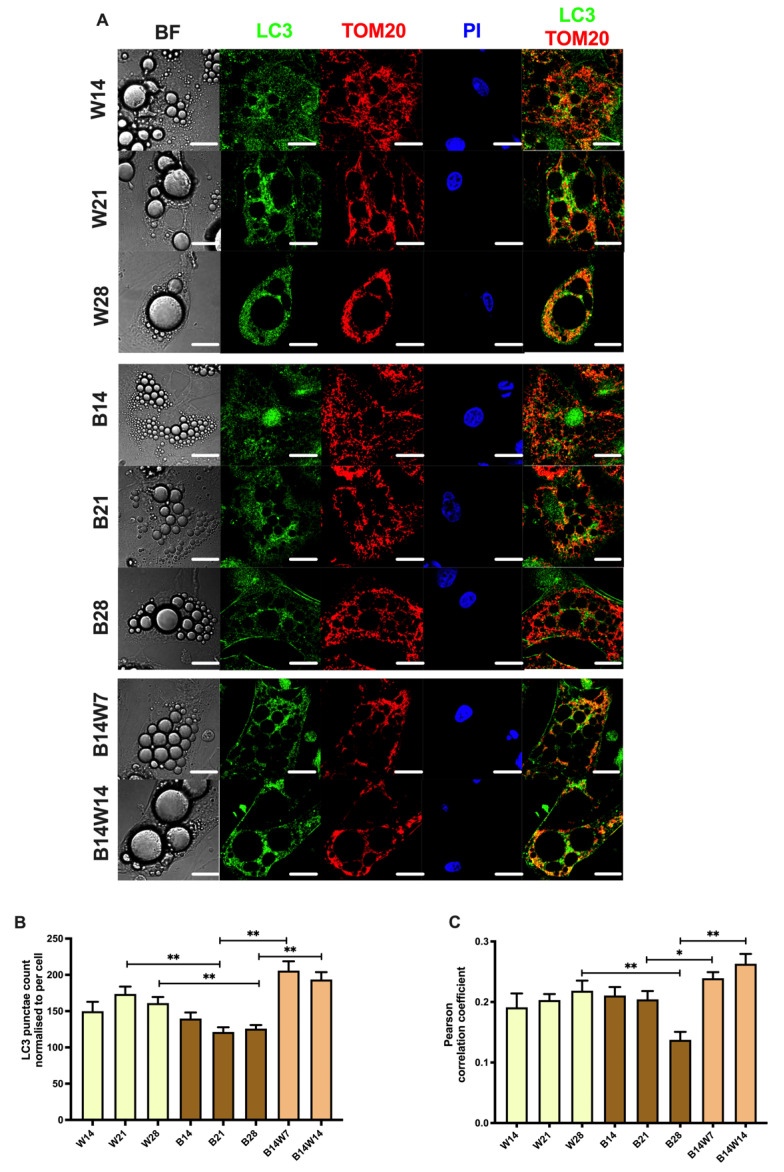
Mitophagy was repressed following long-term rosiglitazone treatment and increased upon beige (B) to white (W) transition. Human primary abdominal subcutaneous preadipocytes were differentiated, as in Figure 1, Figure 2, Figure 3 and Figure 4. (**A**) Representative confocal microscopy images of LC3 and TOM20 immunostaining; nuclei are labeled with propidium iodide (PI), and BF represents brightfield image; scalebars represent 10 μm. (**B**) Quantification of LC3 punctae normalized to per cell (*n* = 50 cells from three donors), (**C**) Quantification of mitophagy as co-localization of LC3 and TOM20 immunostaining (*n* = 50 cells from three donors). Data are presented as mean ± SD. * *p* < 0.05, ** *p* < 0.01. We used one-way ANOVA with Tukey’s post hoc test.

**Figure 6 pharmaceuticals-15-00363-f006:**
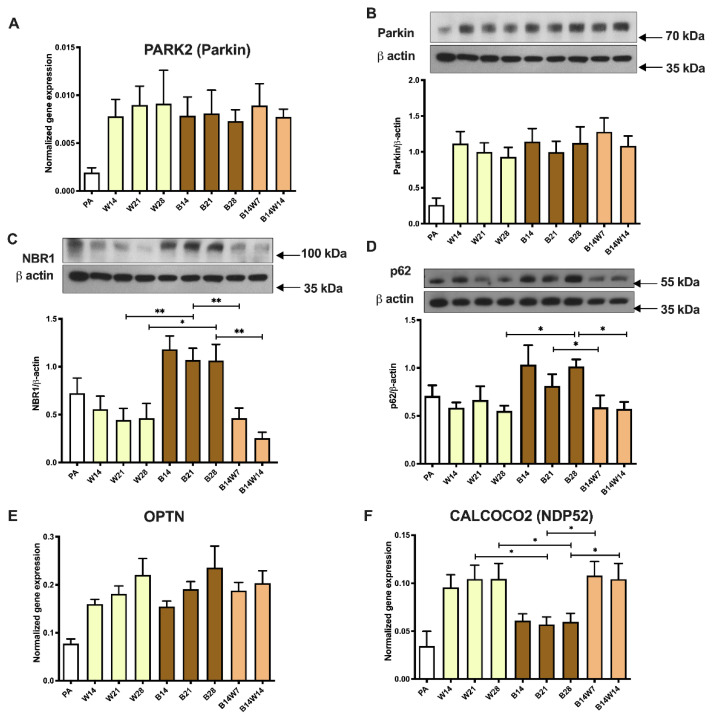
Expression of parkin-dependent mitophagy markers in white (W) and beige (B) adipocytes, and in response to transition. Human primary abdominal subcutaneous preadipocytes were differentiated as in Figure 1, Figure 2, Figure 3, Figure 4 and Figure 5. Quantification of parkin (**A**) gene and (**B**) protein expression. Protein expression of (**C**) NBR1 and (**D**) p62. Gene expression of (**E**) *OPTN* and (**F**) *NDP52*. Gene expression was normalized to *GAPDH* and protein expression to β-actin. Data are presented as mean ± SD. *n* = 6. * *p* < 0.05, ** *p* < 0.01. We performed a Friedman’s test with Dunn’s multiple comparison test (**A**,**B**,**D**) or a one-way ANOVA with Tukey’s post hoc test (**C**,**E**,**F**).

**Figure 7 pharmaceuticals-15-00363-f007:**
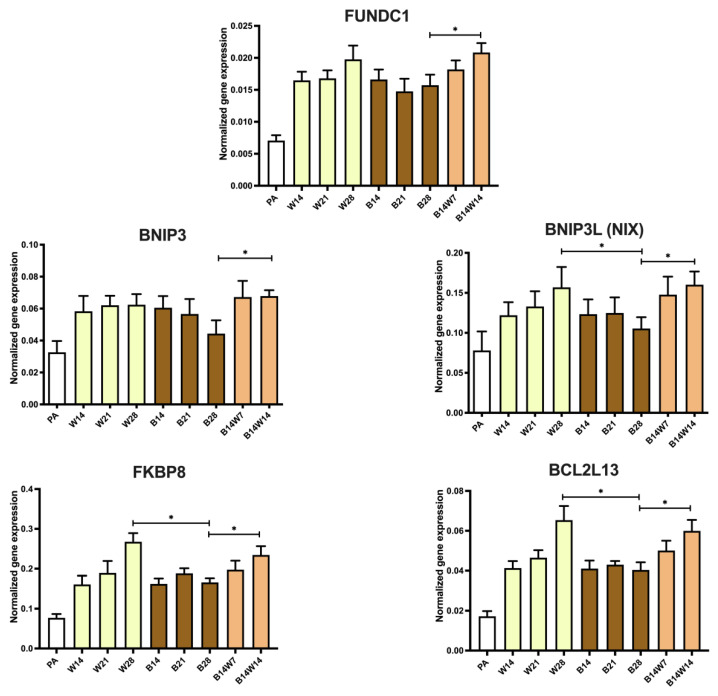
Expression of parkin-independent mitophagy markers in white (W) and beige (B) adipocytes, and in response to transition. Human primary abdominal subcutaneous preadipocytes were differentiated as in Figure 1, Figure 2, Figure 3, Figure 4, Figure 5 and Figure 6. Quantification of gene expression for *FUNDC1*, *BNIP3*, *BNIP3L*, *FKBP8*, and *BCL2L13*. Gene expressions were normalized to *GAPDH*. Data are presented as mean ± SD. *n* = 6. * *p* < 0.05. We used one-way ANOVA with Tukey’s post hoc test.

## Data Availability

The data presented in this study are available on request from the corresponding author. The data are not publicly available due to their containing information that could compromise the privacy of sample donors.

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
