# Peer review of "Mitophagy Mediates the Beige to White Transition of Human Primary Subcutaneous Adipocytes Ex Vivo"

_pharmaceuticals, 2022, doi:10.3390/ph15030363_

Round 1
Reviewer 1 Report
The study concerns the molecular mechanism occurring during beige to white transition of preadipocytes. Authors showed beige differentiation of human primary subcutaneous abdominal preadipocytes, induced by stimulation with PPAR g was successful and manifested by upregulation of UCP1 both at mRNA and protein level. The beige differentiation was also associated with increased mRNA expression for CIDEA. At the same time, indicated markers were expressed constantly at a moderate level in white adipocytes. The authors also showed that differentiation of preadipocytes into beige leads to mitochondrial fragmentation that accompanied by increased UCP-1 expression reflects the thermogenic competency of the cells. In turn in white adipocytes, the organelles remained elongated. Moreover, authors showed OCR raise in beige adipocytes, which collectively with mtDNA accumulation and mitochondria fragmentation suggested inhibition of mitophagy. The Authors performed a profound study of selective autophagy of mitochondria indicating that it is enhanced during beige to white transition. Furthermore, the Authors showed that the parkin-independent pathway may have a significant role in terms of beige to the white transition of human subcutaneous adipocytes.
The article is performed. The introduction section is short and comprehensive. The results were described in detail. The materials and method section is well-organized and protocols are well described. The discussion section is complete, every result was deeply considered by the authors and discussed with current knowledge.
The study brought a closer understanding of the essential molecular events that trigger the entry into beige to white transition. This is particularly important in terms of the development of new therapies e.g. in terms of improving the metabolism of obese patients.
I have two suggestions:
- In my opinion, the authors could easily improve the analysis of mitochondria morphology and network, i.e. performing analysis using MicroP software or 3D reconstructions (please see: e1002212, doi:10.1371/journal.pcbi.1002212). Please add the results of such analysis.
- Authors showed that mtDNA is increased – what about analysis of mitochondrion? Authors tested PGC1a, what about other markers ubiquinol-cytochrome c reductase core protein 2 (UQCRC2), transcription factor A, mitochondrial (TFAM), pseudouridylate synthase-like 1 (PUSL1 ), NADH: ubiquinone oxidoreductase subunit A9 (NDUFA9), cytochrome c oxidase subunit 4I1 (COX4I1), PIGB opposite strand 1 (PIGBOS1), mitochondrial ribosomal protein L24 (MRPL24), mitochondrial transcription termination factor 4 (MTERF4), mitochondrial inner membrane protein (OXA1L) ?
Author Response
The study concerns the molecular mechanism occurring during beige to white transition of preadipocytes. Authors showed beige differentiation of human primary subcutaneous abdominal preadipocytes, induced by stimulation with PPAR g was successful and manifested by upregulation of UCP1 both at mRNA and protein level. The beige differentiation was also associated with increased mRNA expression for CIDEA. At the same time, indicated markers were expressed constantly at a moderate level in white adipocytes. The authors also showed that differentiation of preadipocytes into beige leads to mitochondrial fragmentation that accompanied by increased UCP-1 expression reflects the thermogenic competency of the cells. In turn in white adipocytes, the organelles remained elongated. Moreover, authors showed OCR raise in beige adipocytes, which collectively with mtDNA accumulation and mitochondria fragmentation suggested inhibition of mitophagy. The Authors performed a profound study of selective autophagy of mitochondria indicating that it is enhanced during beige to white transition. Furthermore, the Authors showed that the parkin-independent pathway may have a significant role in terms of beige to the white transition of human subcutaneous adipocytes.
The article is performed. The introduction section is short and comprehensive. The results were described in detail. The materials and method section is well-organized and protocols are well described. The discussion section is complete, every result was deeply considered by the authors and discussed with current knowledge.
The study brought a closer understanding of the essential molecular events that trigger the entry into beige to white transition. This is particularly important in terms of the development of new therapies e.g. in terms of improving the metabolism of obese patients.
Answer: We thank the Reviewer for the positive comments.
I have two suggestions:
- In my opinion, the authors could easily improve the analysis of mitochondria morphology and network, i.e. performing analysis using MicroP software or 3D reconstructions (please see: e1002212, doi:10.1371/journal.pcbi.1002212). Please add the results of such analysis.
Answer: As it was previously described by our research group (Ref. 37, 46), TOM20 immunostaining images were converted to binary form followed by processing with FIJI. Fragmented mitochondria were analyzed from the binary TOM20 immunostaining images with size (pixel2) 0–100 AU and circularity 0-1 AU. The optimum size values for fragmented mitochondria were determined based on analysis of all immunostaining images and manual verification of the counting accuracy by checking the outlines of counts. Fragmented mitochondria content were normalized to per nucleus for individual image. We have amended 4.5. with the more detailed description of image analysis in the revised version of the manuscript.
- Authors showed that mtDNA is increased – what about analysis of mitochondrion? Authors tested PGC1a, what about other markers ubiquinol-cytochrome c reductase core protein 2 (UQCRC2), transcription factor A, mitochondrial (TFAM), pseudouridylate synthase-like 1 (PUSL1), NADH: ubiquinone oxidoreductase subunit A9 (NDUFA9), cytochrome c oxidase subunit 4I1 (COX4I1), PIGB opposite strand 1 (PIGBOS1), mitochondrial ribosomal protein L24 (MRPL24), mitochondrial transcription termination factor 4 (MTERF4), mitochondrial inner membrane protein (OXA1L) ?
Answer: Due to the time consuming feature of the presented experimental setting and the restricted amount of primary adipose-derived stromal cells, we triaged our experiments based on sample availability and tested selected markers for thermogenesis, mitochondrial biogenesis, and auto/mitophagy. We consider the valuable suggestions of the Reviewer when follow up experiments of this study are designed.
Reviewer 2 Report
The manuscript ID pharmaceuticals-1621052 mainly studies beige to white transition of human primary subcutaneous adipocytes considering the mitophagy process as responsible for particular molecular events. Please see below a list of comments to the authors:
- Within the introduction should be highlighted how this is an original work and not only the extension of previous research.
- It is not clear the main advantages in the selection of the cells studied in this work in respect to other samples.
- A graphical abstract describing the methodology and the main findings would be welcome to better present the work.
- Better details to see how was was determined the control group in the experiments are requested.
- Several expressions and citations in the discussion section are disjoint and mentioned without a clear link with the analysis of the results here presented.
- The non-monotonic trends in the quantification of the different transitions studied in Figures 6 and 7 must be better explained.
- The mitophagy is dependent on different physical mechanisms and interplay conditions that must be distinguished in the discussion section to guarantee its responsibility in the transitions studied. Dynamic of sample temperature and incidence of light can be neglected during experiments within the error bar to make reproducible conclusions? Please argue. You can see for instance https://doi.org/10.1080/21623945.2019.1574194
- Potential applications of this research should be mentioned to contemplate future perspectives for the main findings. The authors are invited to see for instance doi:10.3390/cells9122679
- It is suggested to split the collective citations currently reported in the style of (4-7), and some others throughout the text. This is in order to better justify the importance of each selected reference to be part of this work.
- The list of reference must be updated in several cases and in some cases depurated.
Author Response
The manuscript ID pharmaceuticals-1621052 mainly studies beige to white transition of human primary subcutaneous adipocytes considering the mitophagy process as responsible for particular molecular events. Please see below a list of comments to the authors:
Answer: We are grateful to the Reviewer for the insightful remarks which helped us to interpret our data more correctly.
- Within the introduction should be highlighted how this is an original work and not only the extension of previous research.
Answer: We are aware that our study is a follow up of previous ones, including ours, all of which were introduced and discussed in the manuscript text. We have extended the last paragraph of the Introduction in the revised version of the manuscript highlighting the novelty of our study. We do believe that our current data can be an important addition to the literature with regard to human adipocyte biology.
- It is not clear the main advantages in the selection of the cells studied in this work in respect to other samples.
Answer: Although abdominal subcutaneous adipose tissue of human adults is not highly enriched in thermogenic adipocytes, it contains progenitors that can give rise to beige cells. Due to its relatively easy accessibility, we isolated adipose-derived stromal cells of stromal-vascular fractions of abdominal subcutaneous fat. Protocols for induction of white and beige adipogenesis of subcutaneous progenitors were available; therefore we decided to apply these types of cells to study beige to white adipocyte transition. The advantages and disadvantages of our model, as compared to others, are discussed in the first two paragraphs of the Discussion.
- A graphical abstract describing the methodology and the main findings would be welcome to better present the work.
Answer: A graphical abstract has been provided to the manuscript.
- Better details to see how was determined the control group in the experiments are requested.
Answer: In this study, we cannot categorize groups to control and treated ones. Comparisons for fully differentiated white, beige, and cells that underwent beige to white transition were displayed on the respective days (14, 21, or 28) of adipogenic differentiation.
- Several expressions and citations in the discussion section are disjoint and mentioned without a clear link with the analysis of the results here presented.
Answer: Although the Reviewer has not specified what to amend, the Discussion section has been modified at several points in the revised version of the manuscript and statements that are not closely linked to the current study have been removed.
- The non-monotonic trends in the quantification of the different transitions studied in Figures 6 and 7 must be better explained.
Answer: Expression trends are now discussed in more detail in 2.4. and 2.5 of the revised version of the manuscript.
- The mitophagy is dependent on different physical mechanisms and interplay conditions that must be distinguished in the discussion section to guarantee its responsibility in the transitions studied. Dynamic of sample temperature and incidence of light can be neglected during experiments within the error bar to make reproducible conclusions? Please argue. You can see for instance https://doi.org/10.1080/21623945.2019.1574194
Answer: In the presented ex vivo experimental setup, cells were constantly incubated at 5% CO2 and 37°C (we added this information to 4.2) in a standard cell culture incubator. The cells were immediately lysed or fixed after the end of the differentiation/transition for further experiments.
- Potential applications of this research should be mentioned to contemplate future perspectives for the main findings. The authors are invited to see for instance doi:10.3390/cells9122679
Answer: We have improved the last paragraph of Discussion with respect to the future perspectives of our findings.
- It is suggested to split the collective citations currently reported in the style of (4-7), and some others throughout the text. This is in order to better justify the importance of each selected reference to be part of this work.
Answer: References [4-7], [12-14], and [42-44] were split in the revised version of the manuscript.
- The list of reference must be updated in several cases and in some cases depurated.
Answer: We have updated the References in the revised version of the manuscript. Three previous references were omitted and eight references (Ref. 73-79, 90) were included.
Reviewer 3 Report
The MS Mitophagy mediates beige to white transition of human primary subcutaneous adipocytes ex vivo characterizes the cell culture system of human primary adipocytes. Human primary cultures are known to be chimeric, thus the proper characteristic might significantly improve future culture protocols. The entire project looks as designed properly, although I do have some comments listed below.
- Based on what kind of tests Authors decided to use parametric tests instead of non-parametric? In case, data does not have Gauss distribution, please re-analyze the data with non-parametric tests.
- Data does not represent cohort studies, thus please change graph types into dot or scatter plots.
- BF photos are actually cell morphology images and does not have significant input, till not co-analyzed. Did Authors try to figure what are these “bubbles”? Did they consider to stain them against extracellular vesicles/lipid droplets/apoptotic bodies/other markers?
- By which algorithm Authors assessed the fragmented mitochondria? If the calculation did not based on any advance software, please re-calculate images with MINA plug in for Image J.
- Whenever Authors processed the images, please add a description about used software and the rules by which images/objects were classified.
- In my opinion, images do not fallow the findings presented at graphs.
Author Response
The MS Mitophagy mediates beige to white transition of human primary subcutaneous adipocytes ex vivo characterizes the cell culture system of human primary adipocytes. Human primary cultures are known to be chimeric, thus the proper characteristic might significantly improve future culture protocols. The entire project looks as designed properly, although I do have some comments listed below.
Answer: We are grateful to the Reviewer for the constructive comments which helped us to interpret our data more correctly.
- Based on what kind of tests Authors decided to use parametric tests instead of non-parametric? In case, data does not have Gauss distribution, please re-analyze the data with non-parametric tests.
Answer: According to the comments of the Reviewer, we have tested the normality of distribution of the data by Kolmogorov-Smirnov (n=5 or more) or Shapiro-Wilk (n=4) tests. Then, one-way ANOVA with Tukey’s post hoc test was used for multiple comparisons of groups when the data set followed a normal distribution. Friedman's test with Dunn's multiple comparison test was used for multiple comparisons of groups when the data set has not followed a normal distribution. We have amended 4.7., replaced Figures 1 and 6, and indicated the applied statistics in each Figure legend, accordingly, in the revised version of the manuscript.
- Data does not represent cohort studies, thus please change graph types into dot or scatter plots.
Answer: Although we did not perform a cohort study, bar graphs are routinely used to visualize results of normalized gene expression detected by quantitative PCR, optical density values of immunoblotting, or derivate data of image analysis. Using dot or scatter plots would lead to overcrowded figures, thus more difficult interpretation of the results.
- BF photos are actually cell morphology images and does not have significant input, till not co-analyzed. Did Authors try to figure what are these “bubbles”? Did they consider to stain them against extracellular vesicles/lipid droplets/apoptotic bodies/other markers?
Answer: We have included the brightfield images to show that the adipogenic differentiation was successful and the signals arise from intact adipocytes. The “bubbles” are lipid droplets and the presented cells show the characteristic morphology of ex vivo differentiated adipocytes. These objects showed Nile Red positivity as lipid droplets, in our previous study (Ref. 45). Because of this study focused rather on the induction of autophagy, instead of the assembly of lipid droplets, the currently applied cells were not investigated by lipid staining.
- By which algorithm Authors assessed the fragmented mitochondria? If the calculation did not based on any advance software, please re-calculate images with MINA plug in for Image J.
Answer: As it was previously described by our research group (Ref. 37, 46), TOM20 immunostaining images were converted to binary form followed by processing with FIJI. Fragmented mitochondria were analyzed from the binary TOM20 immunostaining images with size (pixel2) 0–100 AU and circularity 0-1 AU. The optimum size values for fragmented mitochondria were determined based on analysis of all immunostaining images and manual verification of the counting accuracy by checking the outlines of counts. Fragmented mitochondria content were normalized to per nucleus for individual image. We have amended 4.5. with the more detailed description of image analysis in the revised version of the manuscript.
- Whenever Authors processed the images, please add a description about used software and the rules by which images/objects were classified.
Answer: We have amended 4.5. with the more detailed description of image classification in the revised version of the manuscript.
- In my opinion, images do not fallow the findings presented at graphs.
Answer: We have included 50 cells to image analysis in each case and selected one for representation. The Reviewer has not specified which images do not represent the findings reported in the manuscript.
Reviewer 4 Report
It is an interesting study to understand the molecular mechanism underlying the PPARg induced beiging process in human primary subcutaneous adipocytes. A set of well-designed experiments were carried out for ex vivo culture of human primary adipocytes under BAT, WAT, or B-W conditions. Cells’ morphology was examined using imaging and gene expression detection, and cell metabolism test to explore the mitochondria change during these adipogenic conditions. Although the overall experimental design was not overwhelmingly technical savvy or cutting-edging, the results presented provide excellent support to the conclusion and careful discussion. This reviewer also enjoyed the clarity and rationale delivered through the informative introduction and discussion sessions.
Just one minor point that it would be very interesting to discuss the shared and/or different downstream events during beiging induced by PPARgamma or beta3-adrenergic agonist. What unique features or actions of sequence would distinguish these two paths?
Author Response
It is an interesting study to understand the molecular mechanism underlying the PPARg induced beiging process in human primary subcutaneous adipocytes. A set of well-designed experiments were carried out for ex vivo culture of human primary adipocytes under BAT, WAT, or B-W conditions. Cells’ morphology was examined using imaging and gene expression detection, and cell metabolism test to explore the mitochondria change during these adipogenic conditions. Although the overall experimental design was not overwhelmingly technical savvy or cutting-edging, the results presented provide excellent support to the conclusion and careful discussion. This reviewer also enjoyed the clarity and rationale delivered through the informative introduction and discussion sessions.
Just one minor point that it would be very interesting to discuss the shared and/or different downstream events during beiging induced by PPARgamma or beta3-adrenergic agonist. What unique features or actions of sequence would distinguish these two paths?
Answer: We thank the Reviewer for the positive comments. Due to the time consuming feature of the presented experimental setting and the restricted amount of primary adipose-derived stem cells, we triaged our experiments based on sample availability and tested only rosiglitazone-driven beige differentiation, which was previously applied by our research group in multiple studies. We have included two paragraphs in the Discussion of the revised version of the manuscript with respect to the features of PPARγ and β-adrenergic-driven adipocyte browning.
Round 2
Reviewer 2 Report
In the reviewed version of the manuscript, the authors have clarified fundamental points about their results, and it is guaranteed that this work is an original research. An attractive topic with useful and solid information has been presented. Then I can recommend this work for publication as it is.
Reviewer 3 Report
No further comments.
This manuscript is a resubmission of an earlier submission. The following is a list of the peer review reports and author responses from that submission.
Round 1
Reviewer 1 Report
Overall this is an interesting set of observations regarding the process of conversion of beige adipocytes to so called “masked” beige adipocytes. Masked beige adipocytes morphologically resemble white adipocytes but retain the capacity for reactivation. The work builds on that of Kajimura and colleagues that shows mitophagy is less active in beige cells than in white cells and that mitophagy is needed to convert beige cells to “masked” beige adipocytes. While Kajimura demonstrated this in mice it was not demonstrated in human cells. The authors had previously demonstrated that mitophagy was highly active in white and masked beige cells and regulated (suppressed) by adrenergic stimuli in human adipocytes.
While the new manuscript presents some interesting findings it feels highly incremental on the authors previous work (Ref 39). The major addition is a change in differentiation protocol to generate a “reversal” of beige to white. The manuscript would have been vastly stronger had either pharmacological or genetic approaches been used to inhibit autophagy and demonstrate that a loss of autophagy and mitophagy prevented the transition from beige to masked beige cells.
A major concern is the statistics used in the paper. The authors use multiple unpaired t-tests for every figure. Given the number of different conditions (8-10) this means huge numbers of t-tests can be run, making a false positive highly likely. It is unlikely that more than half of the significant results would remain if appropriate statistics were used (ANOVAs, repeated measures ANOVA etc, dependent on conditions). As such strong claims are made for small effect sizes. This casts doubt on many of the results (E,g figure 3 A-C and even figure 3D, all of figure 6). Given the importance of whether autophagy is modulated or not and whether it is parkin dependent or not, this is not merely a technical concern.
Equally the seahorse experiments are of further concern. The fact that even the white adipocytes show greater proton leak OCR than basal OCR, even in white adipocytes, suggests some technical optimization is needed.
Overall, while the data are promising it feels both incremental and currently underdeveloped.
Minor points.
More details on the differentiation protocol need to be provided, even as supplemental – the reference for white differentiation was for SGBS cells, this were hASCs. Given the critical nature of the interventions to understanding the phenotype (what was present for the beige differentiation vs what was removed for the “masking”) this should be made clearer.
Figure 1D Texture sum variance? What is it? Are the stats being done on a per donor or per cell basis? It is unclear why this was used rather than droplet size/number/average number per cell etc.
Page 2 line 51 – uncouples the oxphos from ATP generation.
Figure 2 Relies entirely on mtDNA as a marker of a decline in mitochondrial number. Measuring mitochondially encoded transcripts or running the MitoScience mitochondrial complex antibody kit would provide additional evidence.
Author Response
Overall this is an interesting set of observations regarding the process of conversion of beige adipocytes to so called “masked” beige adipocytes. Masked beige adipocytes morphologically resemble white adipocytes but retain the capacity for reactivation. The work builds on that of Kajimura and colleagues that shows mitophagy is less active in beige cells than in white cells and that mitophagy is needed to convert beige cells to “masked” beige adipocytes. While Kajimura demonstrated this in mice it was not demonstrated in human cells. The authors had previously demonstrated that mitophagy was highly active in white and masked beige cells and regulated (suppressed) by adrenergic stimuli in human adipocytes.
While the new manuscript presents some interesting findings it feels highly incremental on the authors previous work (Ref 39). The major addition is a change in differentiation protocol to generate a “reversal” of beige to white. The manuscript would have been vastly stronger had either pharmacological or genetic approaches been used to inhibit autophagy and demonstrate that a loss of autophagy and mitophagy prevented the transition from beige to masked beige cells.
Answer: We are grateful to the Reviewer for the insightful remarks which helped us to interpret our data more correctly. We are aware that our study is a follow up of previous ones, including ours, all of which were introduced and discussed in the manuscript text. We consider inhibiting autophagy during adipocyte differentiation or transition in future studies, which is beyond the scope of the current manuscript.
A major concern is the statistics used in the paper. The authors use multiple unpaired t-tests for every figure. Given the number of different conditions (8-10) this means huge numbers of t-tests can be run, making a false positive highly likely. It is unlikely that more than half of the significant results would remain if appropriate statistics were used (ANOVAs, repeated measures ANOVA etc, dependent on conditions). As such strong claims are made for small effect sizes. This casts doubt on many of the results (E,g figure 3 A-C and even figure 3D, all of figure 6). Given the importance of whether autophagy is modulated or not and whether it is parkin dependent or not, this is not merely a technical concern.
Answer: We have re-evaluated our data and used one-way ANOVA with Tukey’s post hoc test for multiple comparisons of groups (corrected in 4.7). This resulted in changes in the occurrence of differences that reached the level of statistical significance in Figures 1, 3, and 6. The aforementioned figures were replaced; the corresponding legends and manuscript text were amended in the revised version of the manuscript.
Equally the seahorse experiments are of further concern. The fact that even the white adipocytes show greater proton leak OCR than basal OCR, even in white adipocytes, suggests some technical optimization is needed.
Answer: In Figure 2C, proton leak OCR of dibutyril-cAMP stimulated adipocytes was recorded in the displayed experimental setting that is why these values can be higher than of basal OCR. We apologize for the confusing presentation of these data. In the revised version of the manuscript, “Proton leak” was replaced by “Stimulated proton leak” in Figure 2C and the corresponding manuscript text of section 2.2. was amended.
Overall, while the data are promising it feels both incremental and currently underdeveloped.
Answer: Although the concept of beige to white adipocyte transition is not novel, the underlying transcriptional and cell biological changes have not been investigated in human primary cell models so far. The reasons might be that the amount of primary hASC samples is restricted and the complex experiments require quite a long time and therefore they are difficult to be repeated. We do believe that our current data can be an important addition to the literature with regard to human adipocyte biology. Therefore, we hope that the Reviewers will find the improved version of the manuscript acceptable for publication.
Minor points.
More details on the differentiation protocol need to be provided, even as supplemental – the reference for white differentiation was for SGBS cells, this were hASCs. Given the critical nature of the interventions to understanding the phenotype (what was present for the beige differentiation vs what was removed for the “masking”) this should be made clearer.
Answer: Section 4.2. has been more detailed in the revised version of the manuscript.
Figure 1D Texture sum variance? What is it? Are the stats being done on a per donor or per cell basis? It is unclear why this was used rather than droplet size/number/average number per cell etc.
Answer: We are aware that lipid droplets can be quantified and analyzed directly by segmentation of a fluorescent signal. However, this method showed its disadvantage in that the segmentation highly depends on the threshold algorithm as well as on the consistency of the staining. Our collaborators previously showed (ref. 45) that texture parameters were more sensitive in quantitating adipocyte differentiation or at least gave similar results compared to signals of fluorescent lipid staining. Pictorial information may be grasped in spectral, textural or contextual terms. A textural description concerns itself with spatial distribution of tonal variations within an image. The mathematical equations that describe fourteen distinct textural features of images were reported by Haralick et al. (1973; IEEE Trans. Syst. Man Cybern. SMC-3: 610-621.). From among these, we selected texture sum variance as it could well separate the white and beige adipocyte populations in several cellular models (ref. 43, 44, 46). The statistical analysis was performed per cell basis.
Page 2 line 51 – uncouples the oxphos from ATP generation.
Answer: We apologize for the improper statement, which is now rephrased in the revised version of the manuscript.
Figure 2 Relies entirely on mtDNA as a marker of a decline in mitochondrial number. Measuring mitochondially encoded transcripts or running the MitoScience mitochondrial complex antibody kit would provide additional evidence.
Answer: Due to the time consuming feature of the presented experimental setting and the restricted amount of primary hASCs, we triaged our experiments based on sample availability and tested selected markers for thermogenesis, mitochondrial biogenesis, and auto/mitophagy. We consider the valuable suggestion of the Reviewer when follow up experiments of this study are designed.
Reviewer 2 Report
This study described the involvement of mitophagy in the maintenance of beige phenotypes in human adipocytes. Beige to white transition decreased UCP1 expression, mitochondrial fragmentation and cellular bioenergetics. During the transition, mitophagy was induced to remove mitochondria and obtain levels close to the white adipocyte phenotype.
Overall, it’s a well designed study that confirms the implication of mitophagy using a human cell model. It would have been interesting to see a chloroquine treatment during the transition as a proof of concept for maintaining elevated thermogenesis for combatting obesity in a human model, but I understand if this is beyond the scope of this manuscript.
In figure 2, it seems PGC1a is not responsible for regulating the mitochondrial DNA density. Other genes could be assessed by qPCR like POLG, NRF1 and TFAM. They could give a better association with the mtDNA results.
Minor points
Line 184: UCP1 activity is not the only mechanism responsible for the proton leak. Other mechanisms are at play, like the ADP/ATP carrier. This is probably why UCP1 protein decreased by more than 50% during transition (Fig. 1B) but the proton leak was relatively maintained (Fig. 2C). The sentence should be rephrased.
Author Response
This study described the involvement of mitophagy in the maintenance of beige phenotypes in human adipocytes. Beige to white transition decreased UCP1 expression, mitochondrial fragmentation and cellular bioenergetics. During the transition, mitophagy was induced to remove mitochondria and obtain levels close to the white adipocyte phenotype.
Overall, it’s a well designed study that confirms the implication of mitophagy using a human cell model. It would have been interesting to see a chloroquine treatment during the transition as a proof of concept for maintaining elevated thermogenesis for combatting obesity in a human model, but I understand if this is beyond the scope of this manuscript.
In figure 2, it seems PGC1a is not responsible for regulating the mitochondrial DNA density. Other genes could be assessed by qPCR like POLG, NRF1 and TFAM. They could give a better association with the mtDNA results.
Answer: We thank the Reviewer for the positive comments. Due to the time consuming feature of the presented experimental setting and the restricted amount of primary hASCs, we triaged our experiments based on sample availability and tested selected markers for thermogenesis, mitochondrial biogenesis, and auto/mitophagy. We consider the valuable suggestions of the Reviewer when follow up experiments of this study are designed.
Minor points
Line 184: UCP1 activity is not the only mechanism responsible for the proton leak. Other mechanisms are at play, like the ADP/ATP carrier. This is probably why UCP1 protein decreased by more than 50% during transition (Fig. 1B) but the proton leak was relatively maintained (Fig. 2C). The sentence should be rephrased.
Answer: We apologize for the improper statement, which is now rephrased in the revised version of the manuscript.
Round 2
Reviewer 1 Report
I looked at the manuscript. It was not surprising that the the reviewers have not addressed my central concern of a near total lack of novelty with respect to the work, in 11 days. I initially rejected the paper based on the lack of novelty.
Author Response
We are grateful to the Reviewer for the initial remarks based on which we could greatly improve our manuscript. We have indicated in the Conclusions that although the concept of beige to white adipocyte transition is not novel, the underlying transcriptional and cell biological changes have not been investigated in human primary cell models so far. To our knowledge, we could successfully model white and beige differentiation of human abdominal subcutaneous adipocytes up to 28 days and beige to white transition ex vivo for the first time. We do believe that our current data can be an important addition to the literature with regard to human adipocyte biology.